# Effects of Season, Habitat, and Host Characteristics on Ectoparasites of Wild Rodents in a Mosaic Rural Landscape

**DOI:** 10.3390/ani14020304

**Published:** 2024-01-18

**Authors:** Ana Maria Benedek, Ioana Boeraș, Anamaria Lazăr, Alexandra Sandu, Maria Denisa Cocîrlea, Maria Stănciugelu, Niculina Viorica Cic, Carmen Postolache

**Affiliations:** 1Doctoral School in Ecology, Faculty of Biology, University of Bucharest, 050095 Bucharest, Romania; marcela-alexandra.sandu@s.unibuc.ro (A.S.); carmen.postolache@bio.unibuc.ro (C.P.); 2Faculty of Sciences, Lucian Blaga University of Sibiu, 550012 Sibiu, Romania; ioana.boeras@ulbsibiu.ro; 3Faculty of Food and Tourism, Transylvania University of Braşov, 500036 Brașov, Romania; anamaria.gurzau@unitbv.ro; 4Department of Agricultural Sciences and Food Engineering, Lucian Blaga University of Sibiu, 550012 Sibiu, Romania; denisa.cocirlea@ulbsibiu.ro; 5Institute for Interdisciplinary Studies and Research, Lucian Blaga University of Sibiu, 550024 Sibiu, Romania; 6Brukenthal National Museum, Natural History Museum, 550163 Sibiu, Romania; maria.m.stanciugelu@gmail.com; 7Independent Researcher, 335802 Petrila, Romania; niculinaviorica53@yahoo.com

**Keywords:** ticks, fleas, lice, mites, *Microtus arvalis*, *Apodemus agrarius*, prevalence, mean abundance, dilution effect, land use intensity

## Abstract

**Simple Summary:**

Ectoparasites such as ticks, mites, fleas, and lice are ubiquitous in nature and parasitise a plethora of species ranging from small rodents to large mammals and even humans. Besides their parasitic effects, they also act as vectors for various infectious agents affecting the hosts. Parasite-borne infectious diseases have been and remain a great concern for animal and human health. To better understand the risk of parasite-borne disease, it is imperative to have a good understanding of ectoparasite ecology. This study aimed to test the effects of host species characteristics and the features of the environment inhabited by the hosts on the prevalence and abundance of ectoparasites in rodents from a mostly traditional agricultural mosaic landscape. Results show that the sex, age, and body weight of the host affect the prevalence and abundance of ectoparasites. Contrary to other studies, in our study area, land use intensity had a negative effect on all parasite community parameters.

**Abstract:**

Despite the large number of studies on rodent ectoparasites—most of them vectors of epidemiologically important pathogens—infestation patterns remain poorly understood in various ecological contexts, such as the highly patchy agricultural landscapes. We aimed to relate the infestation of rodents to temporal, habitat, and host variables. We assessed the difference in parasite prevalence and mean abundance depending on host sex, age, and body weight, season, and land use intensity. Furthermore, we analysed the effect of host species abundance and the differential responses of parasites in main and minor host species. The field survey was conducted in a rural landscape in southern Transylvania (Romania) between June and September 2010–2011. We live-trapped small mammals, collected the ticks and fleas, and recorded the presence of lice and mites. Overall, we found the same infestation patterns largely reported in the literature: higher prevalence and mean abundance in heavier adult males, significant seasonality and differences among host species, and evidence of the dilution effect. The uniqueness of our study system was the negative effect of the land use intensity on the prevalence and mean abundance of parasites, explained by the highly patchy mosaic landscape.

## 1. Introduction

Rodents are the most diverse and abundant order of mammals, acting as the main reservoir of zoonotic diseases [1]. In habitats altered by humans through agricultural activities, rodents expose humans to several zoonotic agents circulating in natural ecosystems. Rodent-borne agents can be spread to humans directly, e.g., through bites or after consumption of food or water contaminated with rodent droppings [2], or they can be spread indirectly via parasitic arthropods such as ticks, mites, fleas, and mosquitoes that serve as vectors. These ectoparasites are an integral part of ecological communities and play a paramount role in regulating host populations and shaping host communities [3]. The impact of ectoparasites on an ecosystem or the degree to which they affect humans can only be assessed by analysing the relation to their hosts and how human land disturbances affect their occurrence. 

Given the parasites’ dependence on the host they infect, the prevalence and abundance patterns of parasitic organisms, as well as the dynamics of the relationship between parasites and their rodent hosts, are influenced by a wealth of factors related to host characteristics, such as age, sex, and hormone levels, but also environmental characteristics, such as geographic features, vegetation, and climate or season [4]. These have both a direct effect, important especially for parasite taxa spending most of their time off-host (e.g., ticks), and an indirect effect, mediated by the response—in terms of density, physiological status, behaviour, etc.—of their hosts to the environment. Different parasite taxa may have different preferences for the characteristics of the host and environment [5,6], resulting in contrasting patterns of prevalence and mean abundance, while also shaping the spatio-temporal dynamics of parasite assemblages at the landscape level.

Although the mechanisms governing the rodent host–ectoparasite relationship are intricate and result from the interaction of numerous host- and environment-related factors, there are some prevailing patterns largely—but not universally—reported in the literature: (i) males are more parasitised than females [7,8,9,10]; (ii) older and heavier animals are more parasitised compared with younger, lighter individuals [11,12]; (iii) parasite prevalence and mean abundance show seasonality [13]; (iv) animals living in habitats with woody vegetation have fewer parasites [14,15]; (v) animals in less intensively used habitats have fewer parasites [13]; and (vi) parasite prevalence and abundance vary among host species [16,17].

While most parts of Europe are characterised by intensified agriculture, with extensive monocultures and large-scale use of pesticides and mechanised work, some regions, such as central Romania, still retain traditional farming practices on small plots. This results in a highly heterogeneous landscape with large areas covered by pastures and forest patches and a mosaic of small crop fields, many of which have been abandoned due to the economic inefficiency of this agricultural system [18]. Small mammal communities in these landscapes are diverse and relatively stable, as subpopulations are well connected through ditches and road and field margins, which also, together with fallows, provide refuge during periods of low resource availability on the farmland. Although studies on ectoparasites infesting small mammals are numerous, patterns of the prevalence and abundance of ectoparasites in this group remain poorly understood in various ecological contexts, such as these highly patchy agricultural landscapes. This lack of knowledge has important implications for the health of animals directly affected by parasites [19] and human health.

The main objective of this study was to evaluate and disentangle the effects of habitat and host characteristics on the prevalence and mean abundance of ectoparasite taxa—ticks, mites, fleas, and lice—in rodents in an agricultural mosaic landscape in central Romania, testing some potential mechanisms behind the observed patterns. We assessed whether our study system falls into the patterns usually reported in the literature or whether other (local) mechanisms are involved in the acquisition and propagation of ectoparasites in rodents. We also tested the effect of the abundance of the host population and the whole community, hypothesising that it is negative due to the dilution effect [15], and the mechanisms behind the differential response in main and minor host species—which have significantly higher and lower parasite abundances, respectively. Based on the density-dependent habitat selection theory [20], we hypothesised that, in main host species, parasites would show a weaker response to habitat and host characteristics than in minor host species, and when differences in mean abundance among host species are not significant, parasites would show similar responses.

## 2. Materials and Methods

### 2.1. Study Area

We conducted our study in two localities in southern Transylvania, Romania, between 45°56.190′–46°02.759′ N and 24°27.460′–24°46.040′ E. The position of study sites is depicted in Figure 1. The map was made in QGIS version 3.22.7 [21] using the Google Satellite Hybrid layer available in the software. The study area is part of the special protection area ROSPA0099 Hârtibaciu Plateau, designated under the European Union Directive on the Conservation of Wild Birds. It is a highly patchy rural mosaic landscape (Figure 2) characteristic of southern Transylvania, with small crop (mostly cereals (maize and wheat) and alfalfa) plots interspersed among hayfields and pastures, sometimes overgrazed. Shrub encroachment often occurs in pastures, inducing a secondary succession to broadleaf forests dominated by oak and hornbeam. Many old crops are abandoned; thus, fallows are widespread. They are dominated by tall invasive plant species such as *Solidago canadensis*, *S. gigantea,* and *Erigeron annuus*. Field boundaries are represented by road verges, grassy field margins, and ditches with tall hygrophilous vegetation. The high habitat heterogeneity is associated with a high diversity of plants [22], invertebrates [23], and vertebrates, including small mammals [18].

### 2.2. Rodent Trapping

We live-trapped small mammals using artisanal single-catch plastic box-traps (18 × 8 × 6 cm) (Figure A1) set in transects, including 30 traps placed at intervals of 10 m. Traps were left open for three consecutive days and nights, and we checked them each morning and evening. We used sunflower seeds and apple slices as bait but did not prebait the traps. We conducted the trapping between June and September in 2010 and 2011. In total, we set 82 trap lines. Many traps were disturbed or destroyed by animals or people. Thus, the effective trapping effort was 5571 trap-nights, calculated as the number of traps multiplied by the number of nights, excluding the non-functional traps [18]. Captured individuals were identified based on morphological and biometrical traits according to Aulagnier et al. [24]. They were weighed to the nearest 0.5 g using a 60 g Pesola scale, and sex and age class (adult or subadult) were determined. We used temporary marking by clipping with scissors a 5 × 5 mm spot of fur on the back of the animal [25] to distinguish recaptured individuals. Each animal was released at its trapping site.

### 2.3. Parasite Sampling

In all, we captured 1235 small mammals of 15 species (4 shrews and 11 rodents). In many transects, the densities of host populations were high (especially in autumn), and we did not have time to screen each captured animal for parasites. Therefore, we randomly selected some of them—in all 673—for parasite sampling. Ticks were removed from the skin using tweezers, and fleas were collected by brushing the animal’s fur for two minutes over a white tray. The collected parasites were preserved in 80% ethanol for further identification at the species level using identification keys in Nosek et al. [26] for ticks and Brinck-Lindroth and Smit [27] for fleas. For mites and lice, we noted their presence but did not collect them. We considered the lice present even if we found only eggs on the host hair.

### 2.4. Environment, Host, and Parasite Variables

The focal habitat characteristic in our study was the intensity of land use, which we considered an ordinal variable. We grouped various habitat types into three classes of land use intensity: (1) habitats with a low degree of anthropic use, where the vegetation was natural and undisturbed (forests, forest edges, field margins, and road verges), (2) habitats with intermediate land use intensity, where the vegetation is natural but there is a constant disturbance (pastures), or the disturbance is occasional but with drastic effects on the vegetation, e.g., through mowing (hayfields), or the disturbance is usually absent but the vegetation structure was modified in the recent past (fallows), and (3) intensively used habitats, where the vegetation is artificial and habitat characteristics are regularly altered through tillage and harvesting (crops). The woody vegetation shapes the structure of small mammal communities, affecting various habitat characteristics, such as moisture or food and shelter availability. Because, in our study, open habitats were better represented, in order to keep the data somewhat balanced, we did not distinguish among habitats with various covers of woody vegetation. Therefore, we included the presence of shrubs or trees as a binary predictor in the analyses. There was a partial overlap between the two variables, as all the intensely used habitats had no woody vegetation.

We considered the month of the survey to be an ordinal variable: (1) June, (2) July, (3) August, and (4) September, expecting a linear trend in the monthly changes in the response variables. The year was considered to be a factor. Although we were not particularly interested in the difference between the two years of study, having only two levels (2010 and 2011), it could not be used as a random factor.

As host predictors, we used variables expressed at different levels of the host system: total abundance (community level), species abundance (population level), and taxonomic identity, sex, age, and weight (individual level). Eight species—among them all the shrews—were captured in low numbers (fewer than 10 individuals were examined for parasites) (Table A1). Therefore, for statistical reasons, we excluded them from the analyses, but they were considered in the total abundance of the host community. Abundance was calculated for each transect as the number of trapped individuals (excluding recaptures) per 100 effective trap-nights [18] and was used as a proxy for population and community density.

The data on small mammal populations and community composition and their response to the type of land use and habitat characteristics were previously published [18]. The data on the parasites of rodents in the study area are available in the Appendix A.

For parasites, we calculated the prevalence and mean abundance. Prevalence represents the proportion of individuals in the population that are parasitised, and the mean abundance means the number of parasites per host considering all examined host individuals [28,29]. We calculated the prevalence for each of the four taxa and the mean abundance only for ticks and fleas. We evaluated these parameters for the whole rodent community and separately for the two dominant host species *Microtus arvalis* (common vole) and *Apodemus agrarius* (striped field mouse).

### 2.5. Data Analysis

Prevalence and mean abundance were used as dependent variables. To evaluate their responses to habitat and host characteristics, we used mixed-effects models in the lme4 package [30] in R version 3.6.1 [31], including transect as a random factor. For prevalence, we used the binomial generalised linear mixed models (GLMMs) and, for mean abundance, we used negative binomial GLMMs because overdispersion was significant in all cases except one. Overdispersion was tested using the check_overdispersion function in the performance package [32]. To compare the mixed models and test the effect of the predictors and the random factor in the model, we used the likelihood-ratio test, which assesses the goodness of fit of two competing nested models based on the ratio of their likelihoods. To select the best model we used the stepwise forward selection procedure, starting from the null model—including only the intercept and the random factor—and gradually adding the variables that most increased the model quality (e.g., significance) until adding another predictor did not yield a significantly better model. We tested the zero inflation for each negative binomial model using the function testZeroInflation in package DHARMa [33]. To test the relationship of the parasite taxa with the habitat, time variables, and host species, we used the chi-square test of independence (land use intensity and month were considered in this case to be factors) and, for significant relationships, we evaluated the degree of association between the variables, calculating Cramer’s V statistic using the function CramerV in package rcompanion [34].

To assess the explained variation in the best model and partition it among predictor groups, we used the partR2 function in package partR2 [35]. Confidence intervals were obtained by 2000 bootstrap iterations. For mean abundance, to account for overdispersion, an observation-level random effect was included in the model (as the function does not support negative binomial GLMMs). We illustrated the partitioning of variation in tick and flea mean abundance of the dominant rodent species explained by time, habitat, and host characteristics as Euler (Venn) diagrams, constructed by using the function euler in package eulerr [36].

To assess the difference in tick and flea mean abundance between the two dominant rodent host species, we used the non-parametric Wilcoxon test because of the non-normal data distribution.

## 3. Results

We sampled external parasites from 673 rodent individuals belonging to seven species (Table A1). The two dominant species were *M. arvalis*, with 320 sampled individuals, and *A. agrarius*, with 210 individuals. The two species contributed 78.7% of the total number of captured small mammals.

Collected ticks and fleas were identified to genus or species level. Most collected ticks were larvae (*n* = 729), with fewer nymphs (*n* = 101) and no adults. *Dermacentor marginatus* (*n* = 279) and *Ixodes redicorzevi* (*n* = 278) were most abundant, with *I. redicorzevi* strongly associated with *A. agrarius* (*n* = 190). For some *Ixodes* ticks (*n* = 12), species-level identification based on morphological criteria was not possible. Ticks of genus *Ripicephalus* were also abundant (*n* = 261), with a few individuals identified as *R. sanguineus* (*n* = 7). Among fleas, *Ctenopthalmus* sp. (possibly represented by more than one species) was the most abundant (*n* = 69) and was associated mainly with *M. arvalis* (*n* = 45), as was *Hystricopsylla talpae* (*n* = 10, *n* = 6 in *M. arvalis*). *Megabothris turbidus* (*n* = 55) was the least host-specific, being found on all the examined rodent species. *Palaeopsylla soricis* and *Leptopsylla segnis* were found only accidentally (*n* = 1).

### 3.1. Distribution of Host Species across the Land Use Intensity Gradient

The distribution of host species in habitats with various degrees of land use intensity was significantly different (χ^2^ = 174.5, df = 12, *p* < 0.001, Cramer’s V = 0.348). *Microtus arvalis* was favoured by the moderate land use, which is characteristic of pastures and hayfields, where it was sometimes the only captured species. *Microtus arvalis* reached 65% of the captured individuals in habitats with medium-intensity land use (Table A1). For most species (*Apodemus agrarius, Apodemus flavicollis, Apodemus sylvaticus,* and *Microtus subterraneus*), abundances decreased with increasing land use intensity, but, due to the high variability among habitats with the same land use intensity, the relationships were not statistically significant. *Microtus subterraneus* was captured only in transects with natural or seminatural vegetation, being absent in high-intensity land use habitats. In opposition, *Mus musculus* was characteristic of intensely used habitats, being captured mostly in crops, with only a few individuals found in more natural habitats. In habitats with medium-intensity land use, we recorded the highest number of captured individuals (728, of which 316 were examined for parasites) and species richness (14 species, of which 7 were examined) with the highest trapping effort (Table A1). In low- and high-intensity land use, we deployed a similar trapping effort but abundance (332 versus 175 individuals) and species richness (11 versus 6 species) were almost double in undisturbed habitats compared with the farmland (Table A1).

### 3.2. Variation in Ectoparasite Prevalence

Mites had the highest overall prevalence (0.564, 95% CI = 0.526; 0.6), followed by fleas (0.312, 95% CI = 0.276; 0.349), ticks (0.233, 95% CI = 0.2; 0.264), and lice (0.086, 95% CI = 0.139; 0.197) (Table A2). Arthropod parasite taxa showed a host specificity of medium intensity in the two dominant rodents, given mainly by the significant positive association of ticks with *A. agrarius* (Table A3), which has a more than double tick prevalence (0.319) compared with *M. arvalis* (0.143) (Table A2). Ticks were more prevalent in males (0.286) compared with females (0.184), while mites had a slightly higher prevalence in females. Lice and fleas did not show any preference for the host sex, but the latter had a higher prevalence in adults than subadults (Table A2).

Prevalence varied along the surveyed gradients. All four parasite taxa showed a decreasing trend along the increasing intensity of land use gradient (Table A2), but the association was significant only for ticks (Table A3), whose prevalence decreased from 0.328 in habitats with low-intensity land use to 0.135 in high-intensity land use habitats, and mites, decreasing from 0.609 to 0.327. The presence of woody vegetation was associated (Table A3) with significantly higher prevalences of arachnid parasites (ticks (0.44 in woody vegetation habitats and 0.201 in open habitats) and mites (0.681 and 0.546, respectively)). Insect parasites (fleas and lice) had a higher prevalence in open habitats (Table A2), but the difference was not significant. Prevalence was similar between the two years of study, except for mites, which had a higher prevalence (0.639) in 2010 compared with 2011 (0.52). In contrast, seasonal variation was strong and different among taxa. Tick prevalence decreased strongly from June (0.593) to September (0.136), while mite and flea prevalence increased during the study period from 0.278 to 0.677 in mites and from 0.148 to 0.389 in fleas. The prevalence of lice did not show a seasonal trend (Table A2). However, the monthly variations in prevalence were significant, with a minimum value in June (0.093), followed by the maximum value in July (0.244).

### 3.3. Predictors of Arthropod Parasite Prevalence on Rodent Hosts

The selected predictors poorly explained parasite prevalence in the two dominant rodents except for ticks in *M. arvalis* (R^2^ = 36.5%). The rest of the models had relatively low and comparable coefficients of determination (below 20%, Figure A2). The partitioning of variation explained by the three groups of predictors—time (month and year) of the survey, habitat (land use intensity, presence of woody vegetation), and host (total and species abundance, sex, age, and weight) characteristics—showed that, overall, the host characteristics best explained the variation in parasite prevalence, except for fleas in *A. agrarius*, which showed a strong seasonality (Figure A2).

Land use intensity had a weak negative effect on the prevalence of parasites within the whole rodent community, except for lice (Table 1). Among the dominant species, its effect was significant for fleas in *A. agrarius*. The presence of woody vegetation increased the prevalence of arachnid parasites in *M. arvalis* and overall.

The prevalence of ticks had a linear decreasing trend from June to September, while a positive trend was recorded for mites and fleas, but prevalence did not differ significantly between the two years of the survey (Table 1).

The identity of host species affected the prevalence of all taxa except fleas, but the difference between the two dominant rodents was significant only for ticks (χ^2^ = 17.09, df = 1, *p* < 0.001, prevalence higher in *A. agrarius*). In lice, the large differences in prevalence among species lead to the highest explained variation (0.669). The abundance of host species had no significant effect on the prevalence of any parasite taxa in either of the two dominant rodents. In contrast, the total abundance of the host community had a negative effect on tick prevalence and a positive effect on mites in *M. arvalis* (Table 1).

Among the individual host characteristics, sex affected the prevalence of mites and ticks, but in opposite ways. The prevalence of mites was higher in females of *A. agrarius* and all rodent species and the prevalence of ticks was higher in male *M. arvalis*. Age had a significant effect on lice and fleas, with subadults being less parasitised (Table 1).

With one exception—fleas overall—differences in prevalence among transects were significant.

### 3.4. Predictors of Tick and Flea Mean Abundance

The flea abundance ranged between 0 and 8 parasites per host, with a mean of 0.36 (95% CI = 0.28; 0.44), and the tick mean abundance ranged between 0 and 197, with a mean of 1.8 (95% CI = 1.12; 2.49). Considering only the infested rodents, the mean intensity of fleas was 1.83 (95% CI = 1.58; 2.08), and the mean intensity of ticks was 7.86 (95% CI = 5.06; 10.66).

The selected predictors were able to best explain the tick mean abundance of *M. arvalis*, the explained variation being 0.56 (Table 2). Tick mean abundance was negatively affected by the total abundance of the host community and decreased from June to September, being higher in habitats with woody vegetation and in males. Effects were similar at the level of the host community, where also the effects of land use intensity and host species were significant. In *A. agrarius*, only the effects of the month and total abundance were significant predictors of tick mean abundance. The explained variation was also lower (0.307) (Table 2).

Flea mean abundance was also best explained in *M. arvalis* (0.173). In opposition to ticks, fleas were positively related to the total abundance of the host community and increased from June to September. Still, males also had a higher mean abundance than females. This was the only model in which age significantly affected parasite mean abundance, with subadults having lower values. In *A. agrarius*, flea mean abundance also increased during the survey months, but it responded to the abundance of the host species, not of the whole host community. At the community level, flea mean abundance decreased with the increase in land use intensity, increased from June to September with the host community abundance and host weight, and was higher in males (Table 2).

Tick mean abundance was significantly different among habitats, the effect of the transect as a random factor being highly significant (*p* < 0.001) in all three models (i.e., for the two dominant species and the host community). In contrast, in fleas, the effect of the transect was not significant (*p* > 0.05) in any model.

### 3.5. Differential Responses of Parasites to Extrinsic Factors in Main and Minor Host Species

Ticks had higher mean abundances and were positively associated with *A. agrarius* (mean = 1.82 ticks/host in *A. agrarius*, mean = 0.9 ticks/host in *M. arvalis*, Wilcoxon test: *p* < 0.001), which acts as the main rodent host for ticks in the study area. The fixed effects accounted for 29.8% of the variation in tick mean abundance in *A. agrarius*, while in *M. arvalis* the explained variation was more than double (63%) (Figure 3), their 95% confidence intervals overlapping only slightly (13.6–50% versus 49–82.5%). Host characteristics and temporal variables had similar unique effects on both species, with an important overlap in *A. agrarius*, probably because only total abundance, which increased from summer to autumn, had a significant effect (Table 2).

In *M. arvalis*, the overlap was negligible because sex, which did not vary in time, was the significant predictor among the host characteristics. Habitat factors had a small effect, partially overlapping with the host characteristics (Figure 3) due to rodent abundance variations across habitats.

Contrary to ticks, fleas did not show a significant association with the rodent species (mean = 0.33 fleas/host in *A. agrarius*, mean = 0.45 fleas/host in *M. arvalis*, Wilcoxon test: *p* = 0.73), so they do not have a major host in the area. According to our expectations, the fixed effects accounted for a similar variation in flea mean abundance in the two dominant rodents (51.5% in *M. arvalis* and 42.1% in *A. agrarius*), most of their 95% confidence intervals overlapping (41.5–96.7% versus 33.9–92.5%). Contrary to ticks, in fleas, temporal variations had a significant unique effect in both species (Figure 3).

## 4. Discussion

### 4.1. Effect of Host Characteristics

#### 4.1.1. Sex

Sex had a significant effect on the prevalence only in the arachnid parasites but the opposite in ticks and mites, with males having a significantly higher prevalence of ticks in *M. arvalis* and a lower prevalence of mites in *A. agrarius* and the rodent community (Table 1). We found the expected patterns for the mean abundance (higher in males both for ticks and fleas, in *M. arvalis,* and overall) (Table 2).

Ticks have often been found in larger numbers in males compared with females across many rodent taxa [37,38]. Male-biased parasitism is an intricate phenomenon involving various mechanisms related to interactions between the traits of parasites and their host and the environment [39,40], so there is no universally accepted model for these observations. Still, most often, two hypotheses have been put forward to explain them. Many studies argue in favour of the higher levels of testosterone in males, which, during the breeding season, increase their movement and home-range size [41] and, in turn, lead to a higher abundance of parasites [7,8,42]. In addition, high testosterone levels have been associated with a poor immune response of mammals, allowing for an increase in parasite abundance [9,43,44,45]. However, this pattern is not universal, and, at least in fleas, male bias appears to be more common in European species [46].

In contrast, in parasitic mites, sex bias could not be proven to drive host preference in small mammals in general [47], being concluded that the prevalence of parasitic mites is not related to the bodily features of their mammalian hosts but the climatic and habitat conditions [48,49,50]. This is in opposition to our results, which show a lower prevalence of mites in males (Table 1) and a higher explanatory power of host characteristics compared with habitat or time (Figure A2). The lack of significant patterns or their inconsistency reported in parasitic mites may be explained by the diversity of this group, both in terms of species and higher-ranking taxa, exceeding in biological and ecological heterogeneity other rodent ectoparasite groups. Therefore, we are less likely to find consistent results compared with ticks, fleas, or lice, which are more homogeneous groups.

#### 4.1.2. Age and Body Weight

In accordance with our expectations, prevalence was higher in adults, but the effect of age was not universal, being significant only for some parasite taxa and host species (Table 1 and Table 2). Adult rodents usually have higher mobility in relation to territoriality and reproduction [51,52], increasing the chances of host–parasite encounters and, thus, of acquiring parasites from the environment [7,8]. However, age-related movement patterns are complicated by dispersal, which is more common in young individuals (natal dispersal) and is influenced by intrinsic and extrinsic factors such as life history patterns and resource availability [53]. In addition, young individuals also have a poorer body condition, which makes them less attractive to parasites, who tend to maximise their food acquisition [54]. On the other hand, younger individuals have a lower immune response due to the incompletely developed immune system and, thus, a reduced ability to cope with parasites [38]. The contrasting effects of these mechanisms may explain the lack of age-related patterns of tick infestation in our study system. Fleas are usually acquired from nests or from infested individuals and seldom acquired directly from the environment, while lice are acquired through interindividual transmission. Therefore, their higher infestation rates in adult hosts cannot be attributed to increased mobility. Instead, lower parasite abundances may also be explained through anti-parasitic behavioural activities, such as grooming [54], which are more frequent in subadults, who have a higher degree of sociality and more interactions with the mother and siblings. For mites, age bias could not be proven to drive host preference [47], which is in accordance with our results and may be explained by the contrasting effects of the above-mentioned age-related mechanisms.

Body weight was included, with the expected effect, in one of our models for mean abundance, with a higher weight predicting increased flea abundance at the host community level (Table 2). As space is one of the limiting factors for living organisms, both as such and as a proxy for resource availability, a larger host body size is expected to allow for the coexistence of more parasites [11,12]. In line with this expectation, a larger body size was found to entail more space and niches and, therefore, larger flea assemblages [55].

Body weight was not included in any of the best models for prevalence (Table 1), but, in some cases, weight was a good substitute for age, yielding models of comparable quality, highlighting the strongly confounding effect of the two host characteristics. The overlap between the effect of age and weight, which are positively correlated, was also revealed by other studies [38,43].

Because rodent males are, on average, larger than females, body size could explain the increased abundance in males. Zduniak et al. [39] found higher tick abundances in males, but, when accounting for body mass, the differences between males and females were no longer significant. Therefore, the authors concluded that the larger body mass of males leads to an increased number of parasites and not the sex itself. In our study, we accounted for the effect of body size, so the male-biased tick mean abundance was not related (only) to weight but also to other differences—physiological and behavioral—between males and females. For fleas, the authors mentioned above did not find a difference between sexes, but they did record a bias based on host body mass. This contrasts with the male-biased flea abundance in our study, and the difference may lie in the host species (*A. flavicollis*) in the cited study, which belongs to a different rodent family than *M. arvalis,* where we found the sex bias.

#### 4.1.3. Host Species

In accordance with our hypothesis, host taxonomic identity had a highly significant effect on the prevalence of all parasite taxa except for fleas (Table 1). It also affected tick abundance (Table 2).

Parasitic patterns were consistent between *M. arvalis* and *A. agrarius*. There was no parameter for which they showed opposite responses to the considered host or environment factor. This suggests that the same mechanisms of parasite acquisition and propagation act in the rodent species (at least in the two dominant ones), although their overall prevalence and abundance may be significantly different. Some patterns were significant in only one of the two species (e.g., sex-biased tick and flea abundance in *M. arvalis*, sex-biased mite prevalence in *A. agrarius*), but the effect was weak in most of these cases.

The density-dependent habitat selection theory postulates that, in free-living species at low population densities, animals are usually more selective, becoming more opportunistic when densities increase [20]. We hypothesised that, in external parasites, similar mechanisms may act in host selection and, therefore, in the main host species, which have higher parasite abundances, parasites would show a weaker response to extrinsic factors than in minor host species, which have lower parasite abundances. As a result, the explained variation in mean abundance would be lower for models of the species with significantly higher infestation, where parasites are more opportunistic. On the other hand, in minor host species, the presence of parasites can be expected to be more random, resulting in a lower explained variation value for models of the species with lower infestation. Our results show that a distinction between minor and main hosts was possible only for ticks, which exhibited higher mean abundances on *A. agrarius*. Because host mobility is a key driver of the acquisition of questing ticks from the environment, the difference in tick abundance between the two dominant rodents may be linked to their diet, as granivores, such as *A. agrarius,* are, in general, more mobile than folivores, such as *M. arvalis*, because of the lower availability of seeds compared with green vegetation [52]. The fixed effects accounted for twice the variation in tick abundance in *M. arvalis* than in *A. agrarius*, while the explained variation in flea abundance was similar in the two rodents. These findings are in line with the density-dependent habitat selection theory, suggesting that external parasites may show host-selection mechanisms similar to those exhibited by free-living species for habitat selection.

#### 4.1.4. Abundance of Host Population and Community

Host population density was included in only one model (flea abundance in *A. agrarius*), and it had a positive effect (Table 2). The abundance of the entire rodent community had a positive effect on fleas and mites and a negative effect on ticks (Table 1 and Table 2). Rodent community abundance increases strongly from summer to autumn due to the recruitment following reproduction. Therefore, there is an overlap in the effect of community density and season. However, both are included in most models, meaning that rodent abundance also affects parasites through other mechanisms along with the seasonal variation.

Ticks, as questing individuals, spend most of their time off-host, and rodents acquire them directly from the environment. At high rodent densities, ticks will spread between individual hosts. In contrast, at low densities the ticks are forced to share the few available hosts, hence the negative effect of total rodent abundance. Parasites acquired from nests and burrows or that spread through interindividual host transmission are favoured by dense host populations [19], and this may explain the positive effect we observed in fleas and lice. However, other studies showed that relationships between flea prevalence and abundance in small mammals are either negative or absent [56], which may be attributed to increased social grooming [54].

In our study system, the total abundance of rodent communities was significantly correlated with species richness, the most favourable habitats supporting high numbers of individuals and species [18]. The dilution effect hypothesis states that increased host diversity dampens parasite transmission by limiting competent host availability [57]. Biodiversity may be another factor involved—through parasite dilution—in the reduction, at higher total rodent abundances, in the prevalence and abundance of ticks, but not fleas, as the host species in our study system showed a variation in competence only for ticks. In addition, the dilution effect is expected to vary among species depending on their ecological specialisation. Specialist parasites are less likely to show a dilution effect because they would be diluted only across some hosts or in some habitats [29]. Therefore, dilution effects are less likely to emerge in higher-level taxa, which usually encompass species with various degrees of ecological specialisation.

### 4.2. Effect of Environment

For parasites that spend most of their life off-host, the effect of the environment is direct (affecting distribution, reproduction, and larval survival) and indirect (in terms of the abundance, behaviour, and physiology of hosts) [16,58].

Ticks, fleas, and lice have different degrees of interaction with their host. Most ticks have a three-host life cycle, spending most of their time off-host, with little host–parasite interaction, and sporadic short feeding periods spent on-host, making ticks more susceptible to the influence of the environment. Most flea species are parasitic only as adults, spending a moderate amount of time on-host, in repeated feeding stints. The remaining time is spent usually in the host nest, where reproduction and larval development occur [16]. Since, in the host nest, conditions are relatively stable, the habitat associations of fleas are also moderate. Lice are permanent parasites manifesting a lifelong interaction with the host, which they rely on not only for feeding but for the whole life cycle [59]. Thus, the environment of lice, represented exclusively by the host body, is constant, and the effect of habitat characteristics and climate are only indirect via the influence on the host. Our results confirm the independence of lice prevalence from environmental factors (Table 1 and Figure A2). Still, they failed to show the difference between ticks and fleas in their habitat association since, in both taxa, host characteristics were more important than the environmental ones (Figure 1). A possible explanation for this is the limited habitat features we assessed, ignoring factors such as soil parameters or vegetation cover and structure that have an established effect on ticks [59,60] as well as the climatic variables (mainly temperature and rainfall) [16].

In terms of vegetation characteristics, we considered only the presence of woody vegetation, whose effect, when significant, was positive on arachnid prevalence (Table 1) and abundance (Table 2) in *M. arvalis* and overall. Thus, concerning the effect of woody vegetation, we did not confirm the patterns usually reported in studies such as those in neotropical landscapes, where the reduction in forest cover through deforestation leads to increased rates of parasitism [14,61], probably involving the parasite dilution effect given the higher rodent diversity in natural forests compared with the impoverished rodent communities in habitats affected by deforestation. However, in our landscape, forests were rather species-poor, most often with only one captured species per habitat, while the greatest richness was found in fallows and field margins [18], both being habitats without woody vegetation. Therefore, our results may be, after all, in line with the dilution effect theory. In addition, ticks and mites, during their off-host period, are sensitive to environmental factors, and the canopy of woody plants creates a relatively stable microclimate of favourable conditions, enhancing their survival [62]. For ticks, vegetation structure and the presence of shrubs are also important in questing and finding a host [63].

#### 4.2.1. Land Use Intensity

Contrary to our expectations, land use intensity had a negative effect on rodent parasites, its effect on their prevalence and abundance being weak but significant at the level of the whole rodent host community in all taxa except for lice (Table 1 and Table 2).

Land use is inextricably linked to the distribution and abundance of small mammals, and, consequently, it shapes the structure and abundance of their parasite assemblages [13,64]. Increased human pressure on habitats leads to the loss of biodiversity. When large predators disappear, the numbers of small mammals that they used to regulate—especially those of generalist species, which can withstand the changes in their environment—spiral out of control. This, in turn, influences ectoparasites’ occurrence and abundance patterns [65,66]. A study on the flea and mite abundances of small mammals in the Serengeti National Park and adjacent villages found that the ectoparasite abundance on small mammals was much higher on the agricultural land than in the National Park [13]. On the farming land, the loss of biodiversity—fewer small mammal species—was associated with higher population densities than in the National Park area, indicating a link between land use, biodiversity, and parasite abundance [13], probably involving the dilution effect. In our study area, although species richness and abundance varied considerably among land uses [18], they were independent of land use intensity. This suggests that, in our study system, the effect of land use intensity—the opposite of the one consistently reported in the literature—is related neither to biodiversity nor the density of the host community.

Increased human pressure on habitats also leads to high stress levels in small mammals [67]. Increased parasitic abundances in intensively used habitats are generally attributed to the lower immune responses of rodents [43] caused by higher stress levels. Therefore, we may assume that the small-scale and patchy intensive use in our studied landscape does not increase stress levels in rodents, who, moreover, benefit from the vicinity of the refugia providing them with food and shelter during periods of low resource availability on the farmland. The agricultural work conducted here (tillage, deployment of pesticides, weed removal) also lowers the density of parasites in the environment, reducing the probability of rodents acquiring new parasites (especially ticks) when living on or moving onto the cropland. In addition, the main hosts of adult ticks, represented by both domestic (sheep) and wild large- and medium-sized mammals, are more abundant in low-intensity (forests, field margins) and intermediate-intensity (pastures) land use habitats.

#### 4.2.2. Seasonality

Habitat changes in most areas are seasonal, driven by fluctuations in the temperature and rainfall regime. In addition, host reproduction, accompanied by changes in host behaviour, physiology, and abundance, is also seasonal. Therefore, significant seasonal patterns of parasite infestation are expected, especially among parasites that spend their time both on- and off-host [16], with maximum infestation recorded during the season with the greatest parasite activity [17]. Except for permanent parasites (lice), all parasite taxa showed a linear trend from June to September, decreasing for ticks, increasing for mites and fleas, and stronger for the mean abundance (Table 2) than prevalence (Table 1). The decreasing seasonal trend between June and September in mites and fleas may result from bivoltinism [68], our study period ranging from the decline following the first peak in spring to the second peak in autumn. The opposite trend in ticks may be explained by the high overall rodent density in the study area, which allows most tick larvae to feed shortly after they hatch in spring.

Contrary to the strong seasonal patterns, the differences between the two years of the study in prevalence and mean abundance were not significant in any of the parasite taxa, the year-to-year stability of the parasitic community being probably promoted by the stability of the rodent host community, which did not show significant changes in abundance or species composition [18].

### 4.3. Disentangling the Spatial, Temporal, and Host Effects

None of the models for parasite abundance had significant zero-inflation. In addition, the effects of the considered predictors were generally consistent and stronger for abundance than for prevalence (Table 1 and Table 2). This suggests that similar mechanisms are involved in the acquisition and proliferation of parasites in our study system.

Overall, host characteristics best explained the variation in parasite prevalence, except for fleas in *A. agrarius*, which showed a strong seasonality (Figure A2), with similar patterns found for tick and flea abundance (Figure 3). This contrasts with the results of large-scale studies that reveal the important role of the environment in the host–ectoparasite relationship [47] and show that the host environment, rather than host characteristics, affects parasitic mites [69] and fleas [47], and it probably represents an artifact created by the limited habitat characteristics included in the models as explained above. Expanding these models by including a wider range of host environment descriptors would create a better image of the patterns describing the ectoparasite assemblage in our study area. Disentangling the effect of various intrinsic and extrinsic factors on the parasite prevalence and abundance could offer a deeper insight into the mechanisms governing the parasite–rodent host relationship. This may also have epidemiological implications in predicting the transmission risk of rodent-borne diseases [13,14,17,38].

### 4.4. Limitations, Practical Implications, and Future Research Directions

Firstly, we report results based on analysis of data on ectoparasites at a higher taxon level and, as we mentioned above, the variations in biological and ecological characteristics of species belonging to the same taxa may obscure the mechanisms behind the observed patterns, missing potential confounding factors due to the different characteristics of two or more species belonging to the same group. Secondly, although our study focused on several variables related to host and environment characteristics, it was conducted during a relatively short time span (four months in two consecutive years). Thus, our analysis cannot lead to exhaustive conclusions but rather to the proposal of trends in the ectoparasite–host–environment relationships. Longer and replicated research periods will be needed to confirm the proposed trends. In addition, the month of study, considered in our analysis as the temporal variable, encompasses variations in climatic conditions but also changes in host and parasite populations resulting from their life cycles. To disentangle these effects, future research needs to include local and regional climate variables to establish mechanisms of seasonality of the ectoparasite–rodent host relationship.

In recent years, diseases associated with small mammal reservoirs have increased and research on reservoir ecology has become an integral part of assessing the potential risk to humans and livestock. In Romania, among the four parasite taxa considered in our study, ticks pose an important risk to human health because of the numerous zoonotic pathogens they carry, such as tick-borne encephalitis virus (TBEV), *Anaplasma phagocytophilum*, *Francisella tularensis, Rickettsia* spp., *Babesia* spp., and *Borrelia* spp. [70], the latter causing Lyme boreliosis, which is currently affecting many people across the country, with 142 new cases confirmed in 2021 [71]. The importance of understanding host and vector ecology has been recognised in bacterial, parasitic, and viral diseases [72]. For a high degree of effectiveness of control measures, knowledge of the ecology of parasite populations and their relationships with different environmental variables, or host populations, is required. To enhance its value for public health management, future research should also focus on questing ticks, which pose the highest risk to people and livestock, and their screening for epidemiologically important pathogens.

## 5. Conclusions

In the current study on assemblages of rodent ectoparasites, we found similar patterns of parasite abundance and prevalence to those reported in the literature, mostly consistent in the two dominant rodent species *Microtus arvalis* and *Apodemus agrarius*. The uniqueness of our study system is represented by the negative effect of land use intensity on almost all the considered parameters of the parasite assemblages. These results contrast with the strong and significantly positive effect of land use intensity reported from agricultural areas in other parts of the world. This stark contrast may be explained by the specific landscape pattern in our study area, which is characterised by a highly patchy mosaic of plots with various agricultural land uses or forests. This landscape provides close-by refugia for rodent populations during periods of low resource availability in crops. Furthermore, this unique landscape supports an abundant and diverse small mammal community and curbs parasite infestations through the dilution effect. Our results call for the preservation of the traditional farming practices and agricultural landscape, which not only ensure the ecological context for a high degree of biodiversity but also reduce for farmers the biomedical hazard posed by rodent- and parasite-borne pathogens.

## Figures and Tables

**Figure 1 animals-14-00304-f001:**
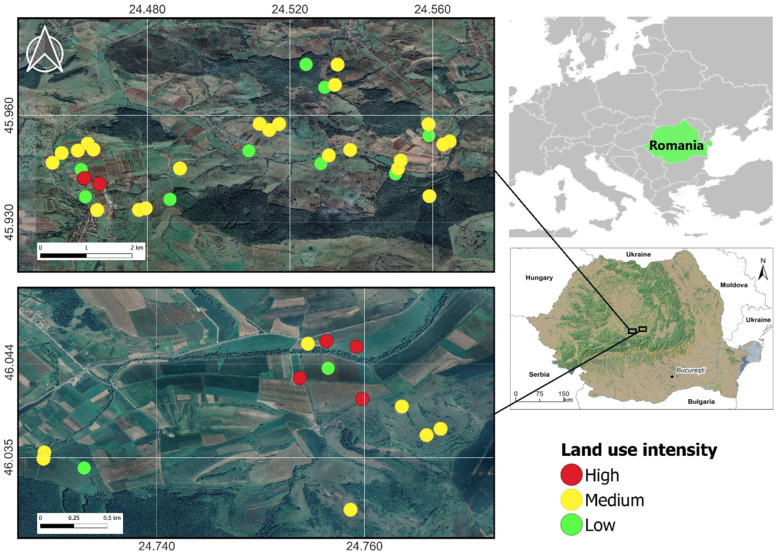
Map depicting the location of the trapping sites (coloured dots) in two rural areas in central Romania.

**Figure 2 animals-14-00304-f002:**
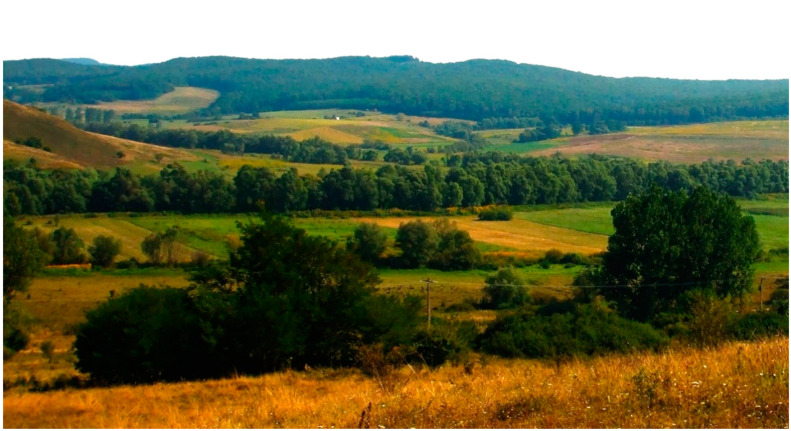
Mosaic agricultural landscape in the study area, characteristic of central Romania (photo, Adriana Vornicu).

**Figure 3 animals-14-00304-f003:**
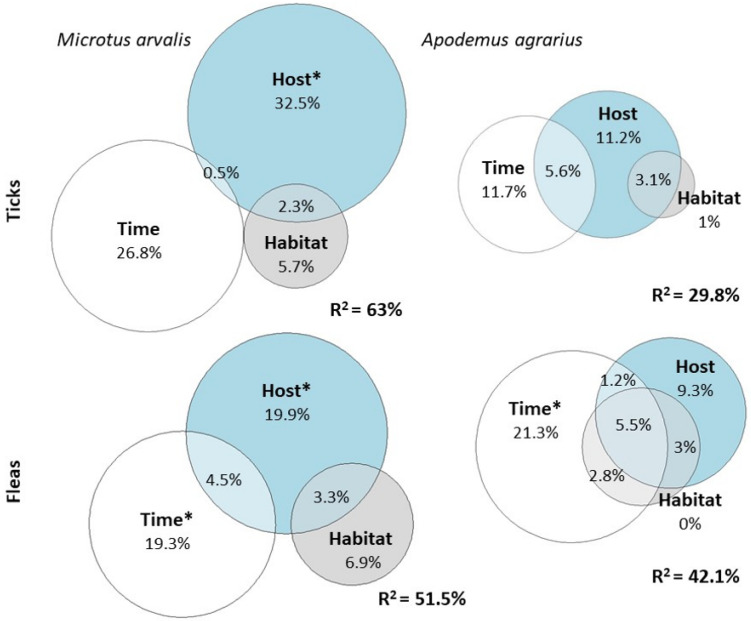
Partitioning of variation in tick and flea mean abundance in the dominant rodent species (*Microtus arvalis* and *Apodemus agrarius*) explained by time (month and year for ticks, month for fleas), habitat (land use intensity, presence of woody vegetation), and host (total and species abundance, age, sex, and weight) characteristics. The unique effect for which 0 was not included in the 95% confidence interval determined by bootstrapping is marked with *. Negative overlap values are considered to be 0.

**Table 1 animals-14-00304-t001:** Explained variation (marginal R^2^, the explained variation accounted for by the fixed effects) and the exponentiated coefficients (Coef.) of the best binomial generalised linear mixed models (GLMMs) for the prevalence of ticks, mites, fleas, and lice in *Microtus arvalis*, *Apodemus agrarius,* and the whole rodent community. The year of the survey, the abundance of host species, and the host weight were not included in any of the best models. Because we used the logit as the link function for the models, coefficients larger than 1 show that the higher values of the predictors increase the odds of infestation, the effect being multiplicative.

Ticks												
Host	*Microtus arvalis*				*Apodemus agrarius*				All rodent host species			
Explained variation	0.267				0.141				0.337			
	Coef.	χ^2^	d.f.	*p*	Coef.	χ^2^	d.f.	*p*	Coef.	χ^2^	d.f.	*p*
Intercept	1.59				9.35				43.9			
Land use intensity									0.6	3.93	1	0.047
Woody vegetation	3.61	4.37	1	0.036					3.52	10.62	1	0.001
Month of survey	0.3	20.43	1	<0.001	0.54	4.42	1	0.035	0.39	23.51	1	<0.001
Total abundance	0.38	7.05	1	0.007	0.98	4.82	1	0.028	0.97	12.42	1	<0.001
Host species	-				-					37.82	6	<0.001
Sex (coef. for males)	3.24	8.18	1	0.004								
**Mites**												
Host	*Microtus arvalis*				*Apodemus agrarius*				All rodent host species			
Explained variation	0.083				0.1				0.158			
	Coef.	χ^2^	d.f.	*p*	Coef.	χ^2^	d.f.	*p*	Coef.	χ^2^	d.f.	*p*
Intercept	0.6				0.21				0.34			
Land use intensity									0.61	4.6	2	0.031
Woody vegetation	3.88	6.5	1	0.013					2.81	7.29	1	0.006
Month of survey					1.98	5.25	1	0.021	1.68	9.7	1	0.001
Total abundance	1.02	5.76	1	0.016								
Host species	-				-					24.86	6	<0.001
Sex (coef. for males)					0.38	9.4	1	0.002	0.63	6.43	1	0.011
**Fleas**												
Host	*Microtus arvalis*				*Apodemus agrarius*				All rodent host species			
Explained variation	0.066				0.142				0.064			
	Coef.	χ^2^	d.f.	*p*	Coef.	χ^2^	d.f.	*p*	Coef.	χ^2^	d.f.	*p*
Intercept	0.14				0.03				0.13			
Land use intensity					0.54	3.99	2	0.046	0.75	4.9	2	0.027
Month of survey	1.53	5.91	1	0.015	2.54	10.94	1	0.001	1.71	28.86	1	<0.001
Age (coef. for subadults)	0.44	9.36	1	0.002					0.51	13.89	1	<0.001
**Lice**												
Host	*Microtus arvalis*				*Apodemus agrarius*				All rodent host species			
Explained variance	0				0.023				0.669			
	Coef.	χ^2^	d.f.	*p*	Coef.	χ^2^	d.f.	*p*	Coef.	χ^2^	d.f.	*p*
Intercept					0.3				0.24			
Host species					-					24.5	6	<0.001
Age (coef. for subadults)					0.39	4.25	1	0.039	0.59	4.87	1	0.027

**Table 2 animals-14-00304-t002:** Explained variation (marginal R^2^, the explained variation accounted for by the fixed effects) and exponentiated coefficient estimates (Coef.) of the best generalised linear mixed models (GLMMs) for the tick and flea abundance of rodent communities and the two dominant species *Microtus arvalis* and *Apodemus agrarius*. The effect of year was tested only for ticks (flea abundance data are only from 2011), but it was not significant. Residuals were modelled using the negative binomial distribution, except for fleas in *A. agrarius*, where overdispersion was not significant and Poisson distribution was used. Because we used log as the link function for the models, coefficients larger than 1 show that higher values of the predictors increase the mean number of parasites, their effect being multiplicative.

Ticks												
Host	*Microtus arvalis*				*Apodemus agrarius*				All rodent host species			
Explained variation	0.56				0.307				0.419			
	Coef.	χ^2^	d.f.	*p*	Coef.	χ^2^	d.f.	*p*	Coef.	χ^2^	d.f.	*p*
Intercept	19.3				31.88				92.75			
Land use intensity									0.56	4.24	2	0.039
Woody vegetation	4.18	5.42	1	0.019					3.17	6.41	1	0.011
Month of survey	0.25	24.72	1	<0.001	0.43	7.96	1	0.004	0.33	26.89	1	<0.001
Total abundance	0.96	9.8	1	0.001	0.98	5.01	1	0.025	0.97	10.57	1	0.001
Host species	-				-					37.57	6	<0.001
Sex (coef. for males)	3.46	12.34	1	<0.001					1.52	4.1	1	0.042
**Fleas**												
Host	*Microtus arvalis*				*Apodemus agrarius*				All rodent host species			
Explained variation	0.173				0.113				0.17			
	Coef.	χ^2^	d.f.	*p*	Coef.	χ^2^	d.f.	*p*	Coef.	χ^2^	d.f.	*p*
Intercept	0.01				0.02				0.01			
Land use intensity									0.66	7.05	2	0.007
Month of survey	2.32	13.47	1	<0.001	2.29	12.28	1	<0.001	1.95	12.73	1	<0.001
Total abundance	1.03	7	1	0.008					1.02	4.8	1	0.028
Species abundance					1.04	6.51	1	0.01				
Sex (coef. for males)	2.2	7.88	1	0.004					1.53	4.45	1	0.034
Age (coef. for subadults)	0.38	7.54	1	0.006								
Host body weight									1.05	24.24	1	<0.001

## Data Availability

The data used for analysis in this study are publicly available as Electronic Appendix A.

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
