# Peer review of "Effects of Season, Habitat, and Host Characteristics on Ectoparasites of Wild Rodents in a Mosaic Rural Landscape"

_animals, 2024, doi:10.3390/ani14020304_

Round 1

Reviewer 1 Report

Comments and Suggestions for Authors

This is a well-conducted research on the ectoparasites of small mammals, carried out in central Europe during the four summer months of two consecutive years.

The Introduction section is well written and documented.

The Material and Methods section is also well written. However, a brief explanation of how Figure 1 has been created and what data has been used for it is missing.

The Results, while comprehensive, are a bit difficult to follow due to the large amount of data reported. Perhaps it could be better to present part of the results, as is already done, using the existing tables 1 and 2, but adding new columns, or creating other new tables.

Concerning the Discussion, the first paragraph is a summary of the objectives and results. It is repetitive and should be removed.

The discussion is well carried out, analyzing the influence on the prevalence and abundance of the taxonomic groups, especially the three most important (ticks, fleas and lice) of the biotic and abiotic variables considered, taking into account their own characteristics. of each group, both its relationship with the host and with the environment and the different types of habitats. Furthermore, differences between the two most abundant host species are analyzed to the extent possible.

The discussion is well conducted with the inclusion of other important and interesting studies for the analysis of the results obtained.

However, one of the limitations of the study, which should be mentioned, is that the analysis of only two years cannot lead to exhaustive conclusions but rather to the proposal of trends in the ectoparasite/host/environment relationships, and that the analysis of a greater number of periods will confirm or not the proposed trends.

Author Response

Comments and Suggestions for Authors

This is a well-conducted research on the ectoparasites of small mammals, carried out in central Europe during the four summer months of two consecutive years.

The Introduction section is well written and documented.

The Material and Methods section is also well written.

We thank the Reviewer for his/her appreciation and for the comments and suggestions that help improving the manuscript. We have tried to address them all in a satisfactory manner. If further changes are needed, we will be happy to make them.

However, a brief explanation of how Figure 1 has been created and what data has been used for it is missing.

In the last paragraph of the Methods, presenting the assessment of the explained variation of the best model and its partitioning among predictor groups, we added:

“We illustrated the partitioning of variation in tick and flea load of the dominant rodent species explained by time, habitat and host characteristics as Euler (Venn) diagrams, constructed by using the function euler in package eulerr (Larsson & Gustafsson 2018).”

The Results, while comprehensive, are a bit difficult to follow due to the large amount of data reported. Perhaps it could be better to present part of the results, as is already done, using the existing tables 1 and 2, but adding new columns, or creating other new tables.

Indeed, due to the extension of the results, this part was difficult to follow. Therefore, we have deleted all the parts of the manuscript that deal with the co-infection. In addition, we moved most of the numerical results from the main text into the existing tables. 

Concerning the Discussion, the first paragraph is a summary of the objectives and results. It is repetitive and should be removed.

We have removed completely the first paragraph of the Discussion.

The discussion is well carried out, analyzing the influence on the prevalence and abundance of the taxonomic groups, especially the three most important (ticks, fleas and lice) of the biotic and abiotic variables considered, taking into account their own characteristics of each group, both its relationship with the host and with the environment and the different types of habitats. Furthermore, differences between the two most abundant host species are analyzed to the extent possible.

The discussion is well conducted with the inclusion of other important and interesting studies for the analysis of the results obtained.

Thank you.

However, one of the limitations of the study, which should be mentioned, is that the analysis of only two years cannot lead to exhaustive conclusions but rather to the proposal of trends in the ectoparasite/host/environment relationships, and that the analysis of a greater number of periods will confirm or not the proposed trends.

Indeed, we acknowledge this limitation of our study and we have added at the end of the Discussion, a new section, entitled “4.4. Limitations, practical implications and further research directions” where we wrote: “Secondly, although our study focused on several variables related to host and environment characteristics, it was conducted during a relatively short time span (four months in two consecutive years). Thus, our analysis cannot lead to exhaustive conclusions but rather to the proposal of trends in the ectoparasite/host/environment relationships. Longer and replicated research periods will be needed to confirm or not the proposed trends. In addition, month of study, considered in our analysis as the temporal variable, encompasses variations in climatic conditions but also changes in host and parasite populations resulting from their life cycles. To disentangle these effects, future research needs to include local and regional climate to establish mechanisms of ectoparasite-rodent host relationship seasonality.”

Reviewer 2 Report

Comments and Suggestions for Authors

The manuscript describes the results of an ecological investigation of the host-parasite relationships between a community of rodent hosts and their ectoparasites, including four taxa (i.e., flea, lice, ticks and mites). The study is interesting and sufficiently innovative, the manuscript is well-written in most of its parts, and the findings are of general interest for an international readership. However, I got some difficulties in reading the results section and my impression is that Authors included too many aspects and types of analysis in their manuscript, making it difficult to clearly understand their findings. Although the system under study is complex, I would suggest to consider the simplification of the overall text, excluding analyses that are not too relevant (e.g., the part on the co-infection). The complexity of the manuscript is recalled by the length of the final part of the Introduction (lines 120-127) where the aims of the study are presented, actually without the identification of a clear research question at the basis of the study.

I have two major concerns on the study design and mostly on the way data are presented.

Firstly, many host species are considered in this study. However, no info on geographical and spatial distribution of hosts is displayed in the manuscript. I can understand that a paper has been already published on this topic (lines 197-200), however the more relevant data that are also needed to interpret the ectoparasite distribution (e.g., species richness and abundance in the three land use intensity areas), should be somehow included also in the present paper, in the main text, or adding a new table.  

Secondly, the considered four taxa, although are internally sharing similar biological characteristics (e.g., permanent parasites vs parasites with environmental phase), may have different ecological features. This aspect is partially mentioned by Authors at lines 463-467, to interpret results on the basis of the difference among ectoparasitic groups in their “biological and ecological heterogeneity”. I think that this point should be more stressed and highlighted as a limitation of the study that is now focused at the higher taxonomic level, missing potential confounding factors due to the different characteristics of two or more species belonging to the same ectoparasitic group. I think that it’s important to add some lines at the beginning of the results to provide at least an overview on the intra-group species diversity, i.e., how many and possibly which species of flea and ticks were collected. This information can be even more interesting if referred to each host species, to provide insights on the generalist or specialist behaviour of the encountered parasitic species (i.e., easily found in different host species, or usually found in only one host).

Finally, I suggest to reflect on the possible practical implications in term of vector-borne disease monitoring and control, coming from the putcomes of the study, besides the vague sentence (lines 689-690) at the end of the Discussion.

Here below some further specific comments.

INTRODUCTION

Line 49: rodents can also be considered wildlife to some extent. Please, amplify to make clearer the concept underpinning this sentence.

Lines 58: I understand Authors’ intention, but the environment and the host are usually considered two different phases in the life cycle of most parasites: better to rephrase.

Lines 70-77: I would shorten this part, since it's similarly presented in the Discussion section at lines 441-449.

Lines 98-99: please rephrase.

Line 102: I would keep this point out of the list, since it refers to all other points.

M&M

Lines 134-137: Consider to include a map of the study area. It may help a lot in interpretation.

Line 146: since artisanal, for a matter of reproducibility, it's important to provide a more detailed description of the trap, possibly including a photo.

Lines 153-154: add brief description and/or appropriate references for rodent identification.

Line 156: some details and references also for fur clipping.

Line 164-166: why collected ectoparasites were preserved in ethanol? Although basilar, provide the morphological characteristics used to differentiate the four taxa. If any further identification of collected parasites was performed, please provide details and references here, and possibly include basic data in the results, as previously suggested.

Lines 194-196: any reference? it's important to use internationally standard methods in density calculation, for a matter of comparison among different studies.

Lines 207-208: not clear. Is it a kind of definition of co-infection? In this case, rephrase.

Data analysis paragraph is well-presented and clear.

RESULTS

Lines 247-248: consider to use the terms mean abundance (mean load in all animals) and mean intensity (mean load in infested animals), as commonly done and internationally proposed. “Load” is fine and generally used, but the suggested terms are more specific in parasite ecology studies (see Bush et al., J. Parasitol., 83(4), 1997 p. 575-583).

Line 251 and following lines: I would suggest to use only the more relevant info, i.e., the p-value and possibly the chi-squared value, deleting other values, throughout the whole Results section, to make easier reading the text.

Line 280: change the words “predict*” to avoid repetition.

Tables 1 and 2: In the first table the heading of the first column is missing. In both tables I suggest to include in the first column the basic explanations that are now provided in the caption of the Table, e.g., “Month (June-September)” or "Sex (males vs. females)".

Line 379: I suggest to change (26) with (n=26). Less ambiguous.

DISCUSSION

Generally, this section is very long and somehow difficult to read, because of the continuous shifts from one level to the other in host types, environment, parasitic taxa and data analysis approaches. It’s difficult to come out with a clear picture of the situation, although Authors identify the general major influence of individual data as one of the main findings of the study. I suggest to shorten and to modify both Results and Discussion sections to improve the clarity and to make it easier for the reader to appreciate the main outcomes of the study.

Line 436: better to use “species” instead of “taxa”, to avoid confusion.

Lines 477-479: this sentence is quite speculative and generic: is it true for all host and parasitic species? any reference to support it? Parasites may have different age-dependent trend, and this is well-known and deeply investigated in endoparasites and to some extents also in ectoparasites (see P. J. Hudson, A. Rizzoli, B.T. Grenfell, H. Heesterbeek and A. P. Dobson, editors. 2002. The Ecology of Wildlife Diseases). The importance of the immune system response in parasite control is surely an important factor.

Line 480: fleas have an environmental phase (larvae): Although they are generally found in the environment usually frequented by the main host, other hosts may accidentally acquire the infection without direct contact. Lice are permanent parasites instead.

Finally, an ethical issue, since the study was conducted manipulating wild animals, that is supposed to be authorised by competent authority or committee. At lines 720-723, it seems that authors were invited in an ongoing trapping activity. However, does this 'invitation' include an ethical assessment? I suggest to include the necessary reference here.

Comments on the Quality of English Language

English form is fine and clear. Some minor changes in few sentences can further improve the manuscript.

Author Response

Comments and Suggestions for Authors

The manuscript describes the results of an ecological investigation of the host-parasite relationships between a community of rodent hosts and their ectoparasites, including four taxa (i.e., flea, lice, ticks and mites). The study is interesting and sufficiently innovative, the manuscript is well-written in most of its parts, and the findings are of general interest for an international readership.

We thank the Reviewer for appreciation and for the comments and suggestions that help improving the manuscript. We have tried to address them all in a satisfactory manner. If further changes are needed, we will be happy to make them.

However, I got some difficulties in reading the results section and my impression is that Authors included too many aspects and types of analysis in their manuscript, making it difficult to clearly understand their findings.

Although the system under study is complex, I would suggest to consider the simplification of the overall text, excluding analyses that are not too relevant (e.g., the part on the co-infection).

We have deleted all the parts of the manuscript that deal with the co-infection. In addition, we moved most of the numerical results from the main text into the tables and deleted the report of non-significant results from the Discussion.  

The complexity of the manuscript is recalled by the length of the final part of the Introduction (lines 120-127) where the aims of the study are presented, actually without the identification of a clear research question at the basis of the study.

We rephrased the introductory part of the last paragraph of the Introduction, to clarify the main objective of the study. It now reads:

“The main objective of the study was to evaluate and disentangle the effects of habitat and host characteristics on the prevalence and abundance of ectoparasite taxa—ticks, mites, fleas and lice—in rodents in an agricultural mosaic landscape in central Romania, and to test some potential mechanisms behind the observed patterns.“

I have two major concerns on the study design and mostly on the way data are presented.

Firstly, many host species are considered in this study. However, no info on geographical and spatial distribution of hosts is displayed in the manuscript. I can understand that a paper has been already published on this topic (lines 197-200), however the more relevant data that are also needed to interpret the ectoparasite distribution (e.g., species richness and abundance in the three land use intensity areas), should be somehow included also in the present paper, in the main text, or adding a new table.  

We have added to the Results section, a subsection, ”3.1. Distribution of host species across the land use intensity gradient”, and completed it with a table in the Appendix.

Secondly, the considered four taxa, although are internally sharing similar biological characteristics (e.g., permanent parasites vs parasites with environmental phase), may have different ecological features. This aspect is partially mentioned by Authors at lines 463-467, to interpret results on the basis of the difference among ectoparasitic groups in their “biological and ecological heterogeneity”. I think that this point should be more stressed and highlighted as a limitation of the study that is now focused at the higher taxonomic level, missing potential confounding factors due to the different characteristics of two or more species belonging to the same ectoparasitic group.

We acknowledge that the main limitation of our study is the approach at the higher taxonomic level, which, due to the variations in biological and ecological characteristics of species belonging to the same taxa may obscure the mechanisms behind the observed patterns in parasite prevalence and abundance. We highlight this in the revised version of the manuscript, in a newly introduced section of the Discussion, entitled “4.4. Limitations and further research directions” where we wrote: “Firstly, we report results based on analysis of data on ectoparasites at higher taxon level and, as we mentioned above, the variations in biological and ecological characteristics of species belonging to the same taxa may obscure the mechanisms behind the observed patterns, missing potential confounding factors due to the different characteristics of two or more species belonging to the same group.  

I think that it’s important to add some lines at the beginning of the results to provide at least an overview on the intra-group species diversity, i.e., how many and possibly which species of flea and ticks were collected. This information can be even more interesting if referred to each host species, to provide insights on the generalist or specialist behaviour of the encountered parasitic species (i.e., easily found in different host species, or usually found in only one host).

We have added at the beinning of the Results section a paragraph including the identified species of fleas, with their host distribution.

Finally, I suggest to reflect on the possible practical implications in term of vector-borne disease monitoring and control, coming from the outcomes of the study, besides the vague sentence (lines 689-690) at the end of the Discussion.

In the newly introduced section “4.4. Limitations, practical implications and future research directions”, we added the following paragraph:

“In recent years, diseases associated with small mammal reservoirs have increased and research on reservoir ecology has become an integral part of assessing potential risk to humans and livestock. In Romania, among the four parasite taxa considered in our study, ticks pose an important risk for human health because of the numerous zoonotic pathogens they carry, such as tick-borne encephalitis virus (TBEV), Anaplasma phagocytophilum, Francisella tularensis, Rickettsia spp., Babesia spp. and Borrelia spp. [71], the last causing Lyme boreliosis, currently affecting many people across the country, with 142 new cases confirmed in 2021 [72]. The importance of understanding host and vector ecology has been recognised in bacterial, parasitic and viral diseases [73]. For high effectiveness of control measures, knowledge of the ecology of parasite populations and their relationships with different environmental variables, or host populations, is required. To enhance its value for the public health management, future research should focus also on questing ticks, which pose the highest risk for people and livestock, and their screeing for epidemiologically important pathogens.”

In addition, we added at the end of the Conclusion, the following sentence:

“Our results call for the preservation of the traditional farming practices and agricultural landscape, which not only ensure the ecological context for a high biodiversity but also reduce for farmers the biomedical hazard involved by rodent- and parasite-borne pathogens.” 

Here below some further specific comments.

INTRODUCTION

Line 49: rodents can also be considered wildlife to some extent. Please, amplify to make clearer the concept underpinning this sentence.

We have deleted „link people and wildlife”, now the phrase reads as: „rodents expose humans to several zoonotic agents circulating in natural ecosystems”.

Lines 58: I understand Authors’ intention, but the environment and the host are usually considered two different phases in the life cycle of most parasites: better to rephrase.

We have rephrased, now the sentence reads as: “Given the parasites' dependence on the host they infect, the prevalence and abundance patterns of parasitic organisms, as well as the dynamics of the relationship between parasites and their rodent hosts, are influenced by a wealth of factors related to host characteristics, such as age, sex and hormone levels, but also environmental characteristics, such as geographic features, vegetation and climate or season [4].”

Lines 70-77: I would shorten this part, since it's similarly presented in the Discussion section at lines 441-449.

At the suggestion of Reviewer 4, we have completely deleted paragraphs 3, 4 and 5 of the Introduction.

Lines 98-99: please rephrase.

We have rephrased, now the sentence reads as: “ii. older and heavier animals are more parasitized compared with younger, lighter individuals [12,13]”

Line 102: I would keep this point out of the list, since it refers to all other points.

We have rephrased point vi, as we intended it to be different from the previous ones.

M&M

Lines 134-137: Consider to include a map of the study area. It may help a lot in interpretation.

We have added to the M&M section a map indicating the location of the surveyed habitats in the studied landscape.

Line 146: since artisanal, for a matter of reproducibility, it's important to provide a more detailed description of the trap, possibly including a photo.

We have inlcuded the photo of a trap as Appendix.

Lines 153-154: add brief description and/or appropriate references for rodent identification.

We have added the reference for the rodent identification.

Line 156: some details and references also for fur clipping.

We have added some details and a reference for fur clipping. Now the sentence reads as: “We used a temporary marking by clipping with scissors a 5 x 5 mm spot of fur on the back of the animal [25] to distinguish recaptured individuals.”

Line 164-166: why collected ectoparasites were preserved in ethanol?

We have added: “for further identification at species level”

Although basilar, provide the morphological characteristics used to differentiate the four taxa. If any further identification of collected parasites was performed, please provide details and references here, and possibly include basic data in the results, as previously suggested.

We have added the references for the identification keys used for ticks and fleas. However, we think that presenting the morphological characteristics used to differentiate the four parasite taxa is not relevant enough to justify the increase on length of the already bulky text.

Lines 194-196: any reference? it's important to use internationally standard methods in density calculation, for a matter of comparison among different studies.

We have added the reference.

Lines 207-208: not clear. Is it a kind of definition of co-infection? In this case, rephrase.

At the suggestion of Reviewer 1, to simplify the results section, we removed completely all the parts of the manuscript referring to co-infection.

Data analysis paragraph is well-presented and clear.

Thank you. We tried to present as detailed and clearly as possible the methods used for data analysis.

RESULTS

Lines 247-248: consider to use the terms mean abundance (mean load in all animals) and mean intensity (mean load in infested animals), as commonly done and internationally proposed. “Load” is fine and generally used, but the suggested terms are more specific in parasite ecology studies (see Bush et al., J. Parasitol., 83(4), 1997 p. 575-583).

We have replaced “parasite load” with „mean abundance” or „parasite abundance” throughout the manuscript.

Line 251 and following lines: I would suggest to use only the more relevant info, i.e., the p-value and possibly the chi-squared value, deleting other values, throughout the whole Results section, to make easier reading the text.

We have moved most of the numerical results from the main text into the tables to make the text more easy to read.

Line 280: change the words “predict*” to avoid repetition.

We have changed “predicted” with “explained”.

Tables 1 and 2: In the first table the heading of the first column is missing. In both tables I suggest to include in the first column the basic explanations that are now provided in the caption of the Table, e.g., “Month (June-September)” or "Sex (males vs. females)".

We have changed the first column in Tables 1 and 2 and simplified their caption.

Line 379: I suggest to change (26) with (n=26). Less ambiguous.

We have completely deleted the part on co-infection.

DISCUSSION

Generally, this section is very long and somehow difficult to read, because of the continuous shifts from one level to the other in host types, environment, parasitic taxa and data analysis approaches. It’s difficult to come out with a clear picture of the situation, although Authors identify the general major influence of individual data as one of the main findings of the study. I suggest to shorten and to modify both Results and Discussion sections to improve the clarity and to make it easier for the reader to appreciate the main outcomes of the study.

We agree that the complexity of our study and our results makes the text difficult to follow, especially by readers who are not very familiar with the subject of our manuscript. In the revised version we deleted some parts of the Results and the corresponding Data analysis and shortened the Discussion, to decrease the information density and increase the flow of writing.

Line 436: better to use “species” instead of “taxa”, to avoid confusion.

We replaced “the two taxa” with “ticks and mites” to clarify the meaning.

Lines 477-479: this sentence is quite speculative and generic: is it true for all host and parasitic species? any reference to support it? Parasites may have different age-dependent trend, and this is well-known and deeply investigated in endoparasites and to some extents also in ectoparasites (see P. J. Hudson, A. Rizzoli, B.T. Grenfell, H. Heesterbeek and A. P. Dobson, editors. 2002. The Ecology of Wildlife Diseases). The importance of the immune system response in parasite control is surely an important factor.

We have rephrased the mentioned paragraph, adding two references. Now it reads as:

“Adult rodents usually have higher mobility in relation to territoriality and reproduction [52,53], increasing the chances of host-parasite encounters and thus of acquiring parasites from the environment [7,8]. However, age-related movement patterns are complicated by dispersal, which is more common in young individuals (natal dispersal) and is influenced by intrinsic and extrinsic factors, such as life history patterns and resource availability [54]. In addition, young individuals also have a poorer body condition, which makes them less attractive for parasites, who tend to maximize their food acquisition [55]. On the other hand, younger individuals have a lower immune response due to the incompletely developed immune system and thus reduced ability to cope with parasites [38]. The contrasting effects of these mechanisms may explain the lack of age-related patterns of tick infestation in our study system.”

Line 480: fleas have an environmental phase (larvae): Although they are generally found in the environment usually frequented by the main host, other hosts may accidentally acquire the infection without direct contact. Lice are permanent parasites instead.

We have rephrased the paragraph, which now reads:

Fleas are usually acquired from nests or from infested individuals and only seldom directly from the environment and lice are acquired through interindividual transmission. Therefore, their higher infestation rates in adult hosts cannot be attributed to increased mobility.”

Finally, an ethical issue, since the study was conducted manipulating wild animals, that is supposed to be authorised by competent authority or committee. At lines 720-723, it seems that authors were invited in an ongoing trapping activity. However, does this 'invitation' include an ethical assessment? I suggest to include the necessary reference here.

We have added the permit number of the ethical committee that approved the working protocol used in the field survey of small mammals.

Comments on the Quality of English Language

English form is fine and clear.

Thank you.

Some minor changes in few sentences can further improve the manuscript.

We have made some changes to increase the flow of our writing. If specific suggestions are made, we will be happy to adhere to them. 

Reviewer 3 Report

Comments and Suggestions for Authors

Authors studied the infestation of four ectoparasitic groups (ticks, mites, lice, and fleas) on two rodent species, the common vole (Microtus avalis) and the striped field mouse (Apodemus agrarius) in southern Transylvania (Romania). Such kind of studies are not novel, but are of importance for the health of animals directly affected by the parasites, but also for human health. 

A special feature of the present manuscript is that the study area is characterized by a highly patchy mosaic of plots with various agricultural land uses or forests, and the uniqueness of the study system is represented by the negative effect of land use intensity on almost all the considered parameters of the parasite assemblages.

The study was carefully planned and carried out, and the results are convincing. The statistical treatment of the data is conclusive, and the manuscript id professionally written. I found only few corrections to be made:

- line 130: replace the full stop by a comma

- References 6, 15 and 19: why here paper titles in uppercase letters?

- References  8 and 9: use the same abbreviation for the journal title in both references.

Author Response

Comments and Suggestions for Authors

Authors studied the infestation of four ectoparasitic groups (ticks, mites, lice, and fleas) on two rodent species, the common vole (Microtus avalis) and the striped field mouse (Apodemus agrarius) in southern Transylvania (Romania). Such kind of studies are not novel, but are of importance for the health of animals directly affected by the parasites, but also for human health. 

A special feature of the present manuscript is that the study area is characterized by a highly patchy mosaic of plots with various agricultural land uses or forests, and the uniqueness of the study system is represented by the negative effect of land use intensity on almost all the considered parameters of the parasite assemblages.

The study was carefully planned and carried out, and the results are convincing. The statistical treatment of the data is conclusive, and the manuscript is professionally written.

We thank the Reviewer for appreciation and for the comments and suggestions.

I found only few corrections to be made:

- line 130: replace the full stop by a comma

We have replaced the full stop by comma.

- References 6, 15 and 19: why here paper titles in uppercase letters?

We have copied the title directly from the article, but indeed, in the reference list they should be in small letters, so we made the change.

- References  8 and 9: use the same abbreviation for the journal title in both references.

We have changed Parasitol. to Parasitology in reference 9.

Reviewer 4 Report

Comments and Suggestions for Authors

Comments on the manuscript “Differential Responses of Rodent Arthropod Parasites to Season, Habitat and Host Characteristics in a Mosaic Rural Landscape” submitted to the Animals

General comments

I appreciate the opportunity to review this exciting manuscript, which is a systematic investigation of the temporal, habitat, and host variables on the tick, flea, mite, and lice parasitism on rodents in Transylvania, Romania.

Based on premises from other studies, the authors investigated whether the findings are in line with what is already known about the relationship between arthropod parasites and rodents, and the possible mechanisms that could explain similarities and differences. Two hypotheses were also tested. Hypothesis 1, due to dilution, there is a negative effect on the abundance of the host population and the whole community; The authors aimed to explain the mechanisms behind the differential response in main and minor host species— which have significantly higher and respectively lower parasite loads. Hypothesis 2, in the most infested rodent species, there would be a weaker response of the parasites to the habitat and characteristics of the host itself, because, at low population densities, animals are usually more selective.

Using traps to capture live animals, the authors carried out an effort of 5571 trap nights of captures. The animals were taxonomically identified, sexed, and weighed. After collecting the parasites, the animals were released. The captures lasted from June to September 2010 and 2011.

Fifteen species of rodents were captured, but some species were in low numbers. Therefore, for statistical reasons, these low-abundance species were excluded from the analyses, but the total abundance of the host community. For the community density estimation, the abundance was calculated for each transect as the number of trapped individuals (excluding recaptures) per 100 effective trap nights.

The prevalence, parasite load and co-infection of parasites, were estimated. All parameters of parasites were calculated for the whole rodent community, but the prevalence was calculated only for ticks and fleas. For the two dominant host species, Microtus arvalis and Apodemus agrarius, there was an emphasis on the statistical analysis.

The data was analyzed with many statistical approaches: mixed effects models, GLMM, zero inflation for each negative binomial model, zero-inflated Poisson models, and so on.

Summarizing the findings and discussion, the infestation patterns are similar to what has already been analyzed in other studies: males are more infested by parasites, partly because they are heavier, together with the putative role of testosterone; there is a significant seasonality and differences among host species. The effect of dilution is evident. However, contrary to common sense in the scientific literature, there was a negative effect of the land use intensity on the prevalence and load of parasites and the number of co-infections, explained by the patchy mosaic landscape.

The manuscript is original and with a topic relevant to the ecology of host-parasite relationship. The study is complex and analyzes many variables with collected data. It is a study with a large sampling effort over four months in two years. Due to these complex characteristics, it is not easy for the reader to understand the theoretical bases, objectives, methods, and findings presented. This requires an additional effort from the authors, which is to write a complex study more concisely and understandably. Although the grammar and lexicon are right, the language seems fragmented, and tedious, which makes reading difficult for readers less familiar with the topic. To improve understanding, I would suggest compartmentalizing the analysis results into tables, facilitating a more fluid and understandable reading.  

Apart from these general comments, I focus on some more detailed aspects of the text, which I present below.

Title

The title must be accurate and complete. As written, the title leads to the ambiguous interpretation that these may be farm animals and not wild rodents. Furthermore, the term "differential responses" also generates ambiguity in a study on the behavior of parasites. I suggest modifying the title.

Simple Summary

Line 27: This phrase is not related to the conclusion of the study and has already served as a justification in the sentence between lines 20 and 22.

Abstract

It is good.

Keywords

Please, note “Instructions for Authors: “Three to ten pertinent keywords need to be added after the abstract”. There are 13 keywords.

Introduction

There are eight paragraphs. It is a very long Introduction. I suggest removing the third, fourth, and fifth paragraphs (lines 69 to 94), which are not essential to understanding the problem, justification, and objective of the study.

Methods

To comply with legal and ethical aspects of the EU, authors need to declare the protocol number and study permission.

Line 137: Characteristic (no typical).

Line 184: Seasonality is an important factor in ecological studies, which is based on defined aspects of climate, photoperiod, etc. The way of presenting the months only using the formal convention of the regular calendar does not seem to contribute much to understanding the temporal choice of the analysis of the abundance and prevalence of parasites. The choice of a temporal analysis, based on the generic definition of "summer to autumn" (e.g., line 399), does not seem accurate. Consider the generic definition of a country's climate (average monthly temperature, etc.), or a macro-region, important aspects of rodent and parasite ecology are lost. Microclimate would be the best approach (see line 539 and [63]). It seems that the temporal analysis in the present study (by months) does not have a good theoretical basis, at least it is not justified by the authors. Furthermore, the climate varies greatly from one year to the next, in many regions of the planet. The time window is short (4 months/two years) to establish "seasonality" in the ecology and generalize the relationship between arthropods and their rodent hosts.

Results

Line 380: There is no discussion to explain this result in M. musculus. What is the explanation for the absence of parasites in most M. musculus individuals?

Line 393 and 412: Why did you apply the Wilcoxon Test? Is the Wilcoxon Test defined in statistical analysis?

Discussion

Line 539: “Density-dependent habitat selection Theory”. How does this Theory relate to the study’s assumptions?

Line 632: Serengeti valley? Serengeti Reserve?

Conclusion

Based on the previous notes, the authors can only evaluate the conclusion after consideration.

References

It is good.

Author Response

Comments and Suggestions for Authors

Comments on the manuscript “Differential Responses of Rodent Arthropod Parasites to Season, Habitat and Host Characteristics in a Mosaic Rural Landscape” submitted to the Animals

General comments

I appreciate the opportunity to review this exciting manuscript, which is a systematic investigation of the temporal, habitat, and host variables on the tick, flea, mite, and lice parasitism on rodents in Transylvania, Romania.

Based on premises from other studies, the authors investigated whether the findings are in line with what is already known about the relationship between arthropod parasites and rodents, and the possible mechanisms that could explain similarities and differences. Two hypotheses were also tested. Hypothesis 1, due to dilution, there is a negative effect on the abundance of the host population and the whole community; The authors aimed to explain the mechanisms behind the differential response in main and minor host species— which have significantly higher and respectively lower parasite loads. Hypothesis 2, in the most infested rodent species, there would be a weaker response of the parasites to the habitat and characteristics of the host itself, because, at low population densities, animals are usually more selective.

Using traps to capture live animals, the authors carried out an effort of 5571 trap nights of captures. The animals were taxonomically identified, sexed, and weighed. After collecting the parasites, the animals were released. The captures lasted from June to September 2010 and 2011.

Fifteen species of rodents were captured, but some species were in low numbers. Therefore, for statistical reasons, these low-abundance species were excluded from the analyses, but the total abundance of the host community. For the community density estimation, the abundance was calculated for each transect as the number of trapped individuals (excluding recaptures) per 100 effective trap nights.

The prevalence, parasite load and co-infection of parasites, were estimated. All parameters of parasites were calculated for the whole rodent community, but the prevalence was calculated only for ticks and fleas. For the two dominant host species, Microtus arvalis and Apodemus agrarius, there was an emphasis on the statistical analysis.

The data was analyzed with many statistical approaches: mixed effects models, GLMM, zero inflation for each negative binomial model, zero-inflated Poisson models, and so on.

Summarizing the findings and discussion, the infestation patterns are similar to what has already been analyzed in other studies: males are more infested by parasites, partly because they are heavier, together with the putative role of testosterone; there is a significant seasonality and differences among host species. The effect of dilution is evident. However, contrary to common sense in the scientific literature, there was a negative effect of the land use intensity on the prevalence and load of parasites and the number of co-infections, explained by the patchy mosaic landscape.

The manuscript is original and with a topic relevant to the ecology of host-parasite relationship.

We thank the Reviewer for appreciation and for the comments and suggestions that help improving the manuscript. We have tried to address them all in a satisfactory manner. If further changes are needed, we will be happy to make them.

The study is complex and analyzes many variables with collected data. It is a study with a large sampling effort over four months in two years. Due to these complex characteristics, it is not easy for the reader to understand the theoretical bases, objectives, methods, and findings presented. This requires an additional effort from the authors, which is to write a complex study more concisely and understandably. Although the grammar and lexicon are right, the language seems fragmented, and tedious, which makes reading difficult for readers less familiar with the topic.

We agree that the complexity of our study and our results makes the text difficult to follow, especially by readers who are not very familiar with the subject of our manuscript. In the revised version we deleted some parts od the results and the corresponding data analysis and discussion, to decrease the information density and increase the flow of writing.

To improve understanding, I would suggest compartmentalizing the analysis results into tables, facilitating a more fluid and understandable reading.  

We have removed most of the numerical results from the Results section and placed them into the existing or new tables. Hopefully the main text is now more easily read.

Apart from these general comments, I focus on some more detailed aspects of the text, which I present below.

Title

The title must be accurate and complete. As written, the title leads to the ambiguous interpretation that these may be farm animals and not wild rodents. Furthermore, the term "differential responses" also generates ambiguity in a study on the behavior of parasites. I suggest modifying the title.

We have changed the title to “Effects of Season, Habitat and Host Characteristics on Ectoparasites of Wild Rodents in a Mosaic Rural Landscape”   

Simple Summary

Line 27: This phrase is not related to the conclusion of the study and has already served as a justification in the sentence between lines 20 and 22.

Indeed, this is redundant, thus we have deleted the last sentence of the summary.

Abstract

It is good.

Thank you.

Keywords

Please, note “Instructions for Authors: “Three to ten pertinent keywords need to be added after the abstract”. There are 13 keywords.

Indeed, we had three extra keywords, so we removed them.

Introduction

There are eight paragraphs. It is a very long Introduction. I suggest removing the third, fourth, and fifth paragraphs (lines 69 to 94), which are not essential to understanding the problem, justification, and objective of the study.

We agree that the introduction is too long, and removed the suggested paragraphs, as they indeed are not essential in the context, and part of the information is found also in the Discussion section.

Methods

To comply with legal and ethical aspects of the EU, authors need to declare the protocol number and study permission.

In the Institutional Review Board Statement we have added the permit number of the ethical committee that approved the working protocol used in the field survey of small mammals.

Line 137: Characteristic (no typical).

We replaced “typical” with “characteristic”.

Line 184: Seasonality is an important factor in ecological studies, which is based on defined aspects of climate, photoperiod, etc. The way of presenting the months only using the formal convention of the regular calendar does not seem to contribute much to understanding the temporal choice of the analysis of the abundance and prevalence of parasites. The choice of a temporal analysis, based on the generic definition of "summer to autumn" (e.g., line 399), does not seem accurate. Consider the generic definition of a country's climate (average monthly temperature, etc.), or a macro-region, important aspects of rodent and parasite ecology are lost. Microclimate would be the best approach (see line 539 and [63]). It seems that the temporal analysis in the present study (by months) does not have a good theoretical basis, at least it is not justified by the authors. Furthermore, the climate varies greatly from one year to the next, in many regions of the planet. The time window is short (4 months/two years) to establish "seasonality" in the ecology and generalize the relationship between arthropods and their rodent hosts.

We ackowledge this limitation of the study, so we added to the section “4.4. Limitations, practical implications and future research directions” the following paragraph:

Secondly, although our study focused on several variables related to host and environment characteristics, it was conducted during a relatively short time span (four months in two consecutive years). Thus, our analysis cannot lead to exhaustive conclusions but rather to the proposal of trends in the ectoparasite/host/environment relationships. Longer and replicated research periods will be needed to confirm or not the proposed trends. In addition, month of study, considered in our analysis as the temporal variable, encompasses variations in climatic conditions but also changes in host and parasite populations resulting from their life cycles. To disentangle these effects, future research needs to include local and regional climate to establish mechanisms of ectoparasite-rodent host relationship seasonality.”

Results

Line 380: There is no discussion to explain this result in M. musculus. What is the explanation for the absence of parasites in most M. musculus individuals?

It is possible that the low rate of parasitism in Mus musculus was the result of this species being found mainly in the corn fields that had no weeds, indicating intensive farming practices (possible deployment of pesticides which may have affected also the parasites). However, we have deleted all paragraphs on parasite co-infection, so now there is no reference on parasites of species other than Apodemus agrarius and Microtus arvalis.

Line 393 and 412: Why did you apply the Wilcoxon Test?

To assess the difference in tick and flea load between the two dominant host species we used the non-parametric Wilcoxon test because the non-normal data distribution.

Is the Wilcoxon Test defined in statistical analysis?

Wilcoxon test was not described in the original Data analysis section, but we included it in the revised version of the manuscript.

Discussion

Line 539: “Density-dependent habitat selection Theory”. How does this Theory relate to the study’s assumptions?

We have added an explanation to clarify how the habitat selection theory relates to the findings of our study. Now the paragraph reads as:

“The density-dependent habitat selection theory postulates that in free-living species, at low population densities, animals are usually more selective, becoming more opportunistic while densities increase [20]. We hypothesized that in external parasites similar mechanisms may act in host selection and therefore, in main host species—which have higher parasite abundances, parasites would show a weaker response to extrinsic factors than in minor host species—which have lower parasite abundances. As a result, the explained variation in mean abundance would be lower for models of the species with significantly higher infestation, where parasites are more opportunistic. On the other hand, in minor host species, the presence of parasites can be expected to be more random, resulting in lower explained variation for models of the species with lower infestation. Our results showed that distinction between minor and main hosts was possible only for ticks—which exhibited higher mean abundances on A. agrarius. Because host mobility is a key driver of acquisition of questing ticks from the environment, the difference in tick abundance between the two dominant rodents may be linked to their diet, as granivores—such as A. agrarius—are, in general, more mobile than folivores—such as M. arvalis—because of lower availability of seeds compared to green vegetation [53]. The fixed effects accounted for twice the variation in tick abundance in M. arvalis than in A. agrarius, while the explained variation in flea abundance was similar in the two rodents. These findings are in line with the density-dependent habitat selection theory, suggesting that external parasites may show host-selecting mechanisms similar to those exhibited by free-living species for habitat selection.”

Line 632: Serengeti valley? Serengeti Reserve?

We added: “Serengeti National Park and adjacent villages”

Conclusion

Based on the previous notes, the authors can only evaluate the conclusion after consideration.

We have revised the Conclusion section, which now reads as:

In the current study on assemblages of rodent ectoparasites, we found similar patterns of parasite abundance and prevalence to those reported in the literature, mostly consistent in the two dominant rodent species, Microtus arvalis and Apodemus agrarius. The uniqueness of our study system is represented by the negative effect of land use intensity on almost all the considered parameters of the parasite assemblages. These results contrast with the strong and significantly positive effect of land use intensity reported from agricultural areas in other parts of the world. This stark contrast may be explained by the specific landscape pattern in our study area, characterized by the highly patchy mosaic of plots with various agricultural land uses or forests. This landscape provides close-by refugia for rodent populations during periods of low availability of resources in crops. Furthermore, this unique landscape supports an abundant and diverse small mammal community and curbs parasite infestations through the dilution effect. Our results call for the preservation of the traditional farming practices and agricultural landscape, which not only ensure the ecological context for a high biodiversity but also reduce for farmers the biomedical hazard involved by rodent- and parasite-borne pathogens.”  

References

It is good.

Thank you.

Round 2

Reviewer 2 Report

Comments and Suggestions for Authors

I appreciate Authors' effort in addressing all points raised in the revision. The manuscript is now acceptable for publication.